# The Multi-Omic Landscape of Primary Breast Tumors and Their Metastases: Expanding the Efficacy of Actionable Therapeutic Targets

**DOI:** 10.3390/genes13091555

**Published:** 2022-08-29

**Authors:** Guang Yang, Tao Lu, Daniel J. Weisenberger, Gangning Liang

**Affiliations:** 1School of Sciences, China Pharmaceutical University, Nanjing 211121, China; 2China Grand Enterprises, Beijing 100101, China; 3State Key Laboratory of Natural Medicines, China Pharmaceutical University, Nanjing 211121, China; 4Department of Biochemistry and Molecular Medicine, University of Southern California, Norris Comprehensive Cancer Center, Los Angeles, CA 90033, USA; 5Department of Urology, University of Southern California, Norris Comprehensive Cancer Center, Los Angeles, CA 90033, USA

**Keywords:** metastatic breast cancer, genomic landscape, tumor microenvironment, DNA methylation alterations

## Abstract

Breast cancer (BC) mortality is almost exclusively due to metastasis, which is the least understood aspect of cancer biology and represents a significant clinical challenge. Although we have witnessed tremendous advancements in the treatment for metastatic breast cancer (mBC), treatment resistance inevitably occurs in most patients. Recently, efforts in characterizing mBC revealed distinctive genomic, epigenomic and transcriptomic (multi-omic) landscapes to that of the primary tumor. Understanding of the molecular underpinnings of mBC is key to understanding resistance to therapy and the development of novel treatment options. This review summarizes the differential molecular landscapes of BC and mBC, provides insights into the genomic heterogeneity of mBC and highlights the therapeutically relevant, multi-omic features that may serve as novel therapeutic targets for mBC patients.

## 1. Introduction

Female breast cancer (BC) is the most diagnosed cancer worldwide, with an incidence of 2.3 million cases and 685,000 deaths in 2020. BC accounts for 30% of all women’s cancers [1]. It is estimated that one in eight women will develop invasive breast cancer in their lifetime, and the odds increase to one in four if there is a first degree relative diagnosed with BC (www.breastcancer.org/facts-statistics, accessed on 22 May 2022). Due to significant progress in screening, diagnosis and systemic adjuvant treatments in early-stage BC, patient survival outcomes have substantially improved, with 5-year survival rates approaching 90% for patients in the USA [2]. Primary breast tumors exhibit metastatic propensity to multiple distant organs, termed metastatic heterogeneity. The most common sites where breast cancer tend to spread are bone, lung, brain, and liver, as well as distal lymph nodes [3]. The 5-year survival rates decline significantly for metastatic BC (mBC) patients, with nearly 23% for patients with bone metastases, 17% for lung metastases and 9% for liver metastases (reviewed in [3]).

Targeted systemic therapies have prolonged median overall survival (OS) [4,5] for mBC patients; however, drug resistance ultimately rises in nearly all patients, thus presenting a major clinical challenge. Approximately 30% of BC patients ultimately succumb to the disease due to metastasis following a variable period of clinical remission [6]. Poor treatment response is also attributed to cancer cell population heterogeneity, which gives rise to treatment-resistant clones [7] and is reflected by the differential expression of prognostic and predictive biomarkers of tumor metastases [8,9].

Interestingly, in vivo studies have demonstrated that tumor metastasis is inherently inefficient, in which most cancer cells fail to arrive at a remote tissue. Furthermore, only a fraction of cancer cells (0.02–0.1%), successfully colonize distant organs and ultimately develop into metastatic tumors [10]. Cancer cells are not under positive selection pressure to metastasize; but rather, hallmarks of metastasis, including increased cell motility, cell invasion, invasion, tumor microenvironment remodeling, cellular plasticity, and colonization of distant sites [11], increase the likelihood of a cancer cell acquiring metastatic potential [12]. Metastatic hallmarks are derived from genetic, epigenetic, and transcriptomic reprogramming; therefore, characterizing the molecular differences between benign and metastatic BC is paramount in prolonging mBC patient survival and advancing the development of novel efficacious therapeutics.

Since cancer evolution can affect treatment strategies, there has been strong interest in investigating the dynamics of metastasis from multiple perspectives, including, but not limited to, lineage tracing [13], genomic and transcriptomic analyses [14,15,16], as well as single-cell gene expression analyses to interrogate cellular heterogeneity. In this review, we summarize the genomic and tumor microenvironment characterizations of primary breast cancer and their metastasis to provide insights into mBC biology, as well as to highlight their clinical relevance and potential actionability.

## 2. Metastatic Breast Cancer Progression Models

Tumor metastasis is a multi-stage evolutionary process that confers a fitness advantage upon cancer cells. Metastasis generally involves several sequential steps (reviewed in [17]) that begin with local invasion, in which cancer cells escape their primary site through stromal cells surrounding the tumor, followed by intravasation, in which the tumor cells invade the blood or lymphatic systems. Tumor cells surviving this step are known as circulating tumor cells (CTCs) [18]. CTCs can escape the circulatory system, enter distal blood vessels, and then invade distal tissues from these blood vessels in a process known as extravasation. Once these cells adapt to the local microenvironment, they are described as disseminated tumor cells (DTCs). DTCs transform into metastatic initiating cells that ultimately form metastatic lesions [3,19]. Key steps to metastasis include the epithelial to mesenchymal transition (EMT) in which tumor cells become invasive and transition into mesenchymal stem cells, and angiogenesis, a process by which the tumor generates blood vessels involving the tumor cells and the local environment.

Based on empirical findings, two models of metastasis have been proposed: the linear model and the parallel model [20,21,22,23]. In the linear model, the development of metastasis occurs relatively late after a subset of clones acquired mutational and growth capabilities that would allow them to survive and grow in distal organs. This model predicts greater genomic similarity between the metastatic and the primary tumors. In contrast, the development of metastasis occurs relatively early in the parallel model, in which the primary and metastatic tumor cells evolve independently. Therefore, owing to selection pressure and the inherent genetics of tumor cells, metastases that arose from the parallel progression model are more genetically divergent (Figure 1). Kroigard and colleagues [23] performed whole exome sequencing (WES) and targeted deep sequencing of 26 sequential tumors from six breast cancer patients. Interestingly, these data support the existence of both the linear and parallel progression models in developing metastases as well as metastasis–metastasis seeding.

Ullah and colleagues [24] performed next-generation sequencing (NGS) of sequentially collected and spatially diverse primary BC tissues, metastases, and auxiliary lymph nodes from mBC patients to identify evolutionary genetic landscapes as a function of metastasis. In agreement with the Kroigard report [23], both linear and parallel progression models of metastasis were evident. In addition, there is support for metastatic spreading and polyclonal seeding, in which metastatic subclones can originate from the primary tumor and other metastases.

## 3. Breast Tumor Immune Microenvironment

Breast cancer progression is controlled by both genetic and epigenetic factors that coordinate crosstalk between tumor cells and their associated immune microenvironment through interactions between immune cells, stromal cells and chemokines [25,26]. The genomic evolution of breast cancer during metastasis has been well elucidated, however, the evolution of immune cells and the tumor microenvironment is less understood. Breast cancer tumor immune microenvironment (TIME) studies have primarily focused on primary tumors [27,28,29]. In early-stage breast cancer, a positive correlation between longer survival and greater tumor infiltration lymphocyte (TIL) count, as well as upregulation of immune-related gene expression, has been observed [30,31]. These features are also associated with greater chemotherapy sensitivity in operable breast cancer patients, evident by a higher pathological complete response rate [32,33]. However, the identities of the specific genomic features associated with high TILs remain elusive.

Recently, numerous studies have profiled immune cell compositions and characterized the differences in immune microenvironment between primary and metastatic breast tumors [34,35,36]. Metastatic tumors have lower immunogenicity, evident by decreased expression of MHC class I and immune proteasome genes coupled with increased expression of HLA-E, as well as the reduction in dendritic cells. Multiple studies have demonstrated that most immune cell types and immune functions were depleted in metastatic lesions (reviewed in [37,38]). Substantially lower TIL counts and PD-L1 expression were also observed in metastatic lesions, possibly due to immune escape. Furthermore, Zhu et al. demonstrated that M2-like macrophages are enriched in metastatic lesions [27]. Their potential role in promoting metastasis warrants further investigation. Interestingly, lymph node metastases have significantly higher PD-L1 expression than visceral metastases [36].

## 4. Histological and Molecular Subgroups of Breast Cancer

### 4.1. Histological Breast Cancer Subgroups

Several histological types of human BC have been reported and characterized (reviewed in [39]). According to the World Health Organization (WHO) database, nearly all BC cases (~95%) are adenocarcinomas that are derived from the terminal duct lobular unit of the breast. The majority of BCs are invasive ductal carcinomas (50–80%). Other classifications include invasive lobular carcinomas (5–15%), mucinous carcinomas (2%), and medullary carcinomas (5–7%) [39,40]. Inflammatory breast cancer (IBC) accounts for ~1% of all BC cases. The earliest presence of breast cancer is ductal carcinoma in situ (DCIS), defined by the presence non-invasive or pre-invasive BC cells that line the breast milk ducts, but have not yet spread into surrounding tissues (www.cancer.org, accessed on 5 June 2022).

BCs are categorized into stages using the American Joint Committee on Cancer TNM system in which the size of the tumor (T), spread to lymph nodes (N) and/or metastases (M) are key defining characteristics. DCIS is commonly referred to as Stage Tis, while Stage 1 diseases are primarily defined as tumors ≤20 mm in size. Stage 2 BCs are 20–50 mm with a spread of up to three lymph nodes, while Stage 3 tumors are >50 mm in size with spread to multiple lymph nodes and tissues surrounding the breast. Finally, Stage 4 BCs are tumors that have metastasized to distal tissues (www.cancer.org, accessed on 5 June 2022).

In addition to TNM staging criteria, the status of steroid hormone receptors (HR), namely estrogen receptor (ER), progesterone receptor (PGR) and human epidermal growth factor receptor 2 (HER2), have become increasingly critical for understanding BC tumorigenesis and aggressiveness (reviewed in [41]). ER and PGR function as transcription factors following their activation by binding to estrogen and progesterone, respectively (reviewed in [42]). In their role as transcription factors, ER and PGR bind to specific DNA elements and chromatin in regulating gene expression of their target sites, thus affecting signaling pathways that involve cell growth and cell function [42]. Most BCs (60–80%) are positive for ER and PGR expression and ER-positive (ER+) BC patients show an improved response to anti-estrogen-based therapies, have longer survival and are less likely to develop recurrent diseases as compared to ER-negative or ER-negative/PGR-negative BC patients [42,43].

ER is expressed in two forms—ER-α (ERα) and ER-β (ERβ). ERα is expressed from the *ESR1* gene, whereas ERβ is expressed from the *ESR2* gene locus. Both ERα and ERβ bind to the estradiol ligand, dimerize and then move to the nucleus where they function as transcription factors to modulate expression of their target genes (summarized in [44]). ERα is thought to have prominent functional roles in breast and ovarian tissues [45] as well as in mammary gland development in mice [46], while ERβ is thought to function in mammary gland epithelium differentiation [47].

HER2 (ErbB2) is a member of the epidermal growth factor family that includes EGFR (HER1), ErbB3 (HER3) and ErbB4 (HER4). HER2 is an oncogene that functions as a transmembrane receptor tyrosine kinase that controls cell growth and differentiation. HER2 is over-expressed or amplified in up to 25% of BCs. HER2 overexpression correlates with poor BC patient outcome, as well as tumor metastasis, tumor aggressiveness [48], and positive lymph node status. HER2-positive (HER2+) BC patients also show poor standard chemotherapy response and = HER2-amplified BC patients display activated PI3K and EGFR (RAS/RAF) signaling and poor outcomes [49].

A subgroup of BC tumors in which ER, PGR and HER2 are negative, termed triple negative breast cancer (TNBC), accounts for 15–25% of all BC cases [50]. TNBCs are not confined to one histological subtype and have diverse molecular profiles; however, it should be noted that TNBCs show expression profiles seen in basal and myoepithelial breast cells [51,52,53]. TNBCs are an aggressive form of human breast cancer, with high rates of metastasis, tumor recurrence, and poor overall patient survival [50]. The mean TNBC patient survival is less than one year, with a dismal 5-year survival rate of only 11% for metastatic TNBC patients [50]. TNBC patients of African American or Hispanic descent have shorter OS compared to other ethnic and racial groups [50,54]. Moreover, younger women, and women of African American or Hispanic descent, are more likely to develop TNBCs, whereas HR+ BCs are more prevalent in older women [55].

### 4.2. Molecular Subgroups of Breast Cancer

In addition to histological BC classification, the large volume of molecular data collected has enabled for classification of molecular BC subgroups based on tumor gene expression profiles: (1) luminal A, (2) luminal B, (3) HER2-enriched and (4) basal-like [51]. Subsequently, Parker and colleagues used 50 genes from a gene expression microarray, termed Prediction-Analysis Microarray 50 (PAM50), to delineate each intrinsic BC molecular subtype [56]. It should be noted that a fifth group, termed normal-like BC, has been reported (reviewed in [52]). Normal-like BCs, as their classifier suggests, show high expression of genes specific to adipose and non-epithelial cells, as well as low expression levels of luminal epithelial cell genes [52]. These poorly understood tumors account for 5–10% of all BCs and showed the best patient prognosis of all five molecular subgroups [57].

Luminal A and B tumors are the most prevalent in ~60% of all BC cases, in which Luminal A cases comprise ~40% of all BCs, while Luminal B cases account for ~20% [58]. The Luminal BCs and are defined by high ER and PGR gene expression, as well as genes involved in ER activation and signaling [58]. Differences in Luminal A and B tumors stem from higher PGR expression and lower expression of cell cycle and proliferation genes in Luminal A tumors [58]. In this regard, the proliferation genes Ki-67 and PCNA are expressed in Luminal B tumors, but not in Luminal A BCs [58]. In addition, Luminal B tumors are enriched for *TP53* somatic mutations, and Luminal B tumors are typically dedifferentiated with higher grade while Luminal A tumors are more differentiated [58]. Finally, Luminal B patients have significantly shorter OS and disease-free survival (DFS) comparted to Luminal A breast cancer patients [58].

HER2 breast tumors account for ~20% of all BCs and are primarily defined by HER2 overexpression, HER2 amplification and low ER expression [58]. In addition, HER2 BCs display amplification and overexpression of key genes surrounding the *ERBB2* locus that include the *GATA4* transcription factor and the GRB7 adaptor protein [52] that binds tyrosine kinases and the RAS oncogene. However, HER2-BCs also display low expression of the GATA3 transcription factor. The anti-HER2 therapy trastuzumab (Herceptin) has dramatically increased HER2-BC patient survival; however, these patients mostly die to disease progression and the development of brain metastases [59].

It should be noted that a new subgroup, termed HER2-low, was more recently described [60,61]. HER2-low BCs are defined as tumors without HER2 amplification, but show weak or intermediate HER2 detection based on immunohistochemistry (IHC), namely IHC 1+ or IHC 2+ status, respectively [61]. HER2-low BC patients comprise nearly 50–55% of all BC cases, are generally ER+, HER2-low tumors are of higher grade and display higher proliferation than HER2-negative tumors [62,63]. HER2-low patients also show a lower immune response than HER2-negative patients [62]. HER2-low patients are resistant to Herceptin therapy. HER2-low BC patients generally have poor outcome [64], as are HER2-low/ER+ BC patients who are over age 55 years old [65]. However, a meta-analysis of the TCGA BC expression data showed that HER2-low patients did not show differences in disease-free survival, progression free survival or overall survival as compared to other BC subgroups [66]. Comparing HER2-low to HER2-negative cases, Denkert and colleagues [67] showed that HER2-low tumors are distinct from HER2-negative tumors with respect to HR status, tumor grade and tumor proliferation. Interestingly, HER2-low patients displayed improved survival over HER2-negative patients as well as differences in neoadjuvant chemotherapy sensitivity. Overall these findings suggest that the clinical aspects of HER2-low BCs are unclear and require further investigation.

Basal-like breast tumors display gene expression profiles akin to basal and myoepithelial breast cells [51,52]. Since TNBCs also show similar gene expression profiles to basal-like BCs, the majority (80%) of TNBCs are indeed basal-like [51,52,53], but display a high level of molecular heterogeneity based on gene expression data [68,69]. Like TNBCs, basal-like tumors lack ER, PGR and HER2 expression. Basal-like BCs also harbor the greatest amount of tumor-specific CNV as compared to the other groups. In addition, basal-like tumors are of higher grade than the other molecular subgroups. Concurrent with this observation, basal-like BCs are aggressive tumors and basal-like BC patients have limited response to conventional therapies, poor outcome and shortened DFS and OS [69,70].

## 5. Genetic Landscape of Breast Cancer

### 5.1. Hereditary Germline Mutations

Most human cancers can develop through the accumulation of somatic mutations of key driver genes that promote cell growth, inhibit DNA repair and bypass cell growth checkpoints to achieve uncontrolled growth and division [71]. However, human cancers can also develop because of hereditary germline mutations from family members. Approximately 5–10% of all BCs have family history of breast or ovarian cancer, and women with germline mutations in breast cancer type 1 (*BRCA1*) and type 2 (*BRCA2*) are of greater risk of developing breast or ovarian cancer [72]. Indeed, *BRCA1* and *BRCA2* mutation carriers have risks of 57% and 49%, respectively, for developing breast cancer [73]. Moreover, *BRCA1* and *BRCA2* mutation carriers have risks of 60% and 55%, respectively, for developing ovarian cancer [73]. However, it should be noted that environmental factors, as well as cellular and allelic heterogeneity are also influential in determining risk of disease [74]. In addition to increased familial BC and ovarian cancer risk due to *BRCA1* and *BRCA2* germline mutations, approximately 40–45% of all hereditary BCs harbor *BRCA1* germline mutations and 35–40% harbor *BRCA2* germline mutations. In total, 80–85% of all hereditary BCs are linked to *BRCA1/2* germline mutations [75,76,77], indicating that these driver mutation events confer substantial advantages in evading growth controls.

BRCA1 and BRCA2 are tumor suppressors that maintain genomic DNA stability by promoting double-strand DNA break (DSB) repair via homologous recombination and non-homologous end-joining (NHEJ) [78]. BRCA1 co-localizes with RAD51 and other DNA repair proteins, and is activated by phosphorylation from the kinases ATM, ATR and CHK2 that are also activated due to DNA damage [78]. Activated BRCA1 functions in G1-S and G2-M cell cycle arrest following ultraviolet or infrared-based DNA damage and acquiring access to specific DNA regions by interactions with histone deacetylases (HDACs) and SWI/SNF chromatin remodelers [78]. Interestingly, BRCA1 also inhibits estrogen-based transcriptional activation of *ESR1* (ER α, ERα) in breast tissues, and studies are suggestive that BRCA1 directly interacts with ERα to inhibit ERα function [79]. Additional germline mutations in breast cancer include *TP53*, *ATM* and *CHK2* [80]. Interestingly, all these genes are involved in DNA repair and genomic integrity.

Beyond *BRCA1* and *BRCA2* genetic variation, several studies have interrogated germline variation of cancer-related genes in BC and mBC patients. A report from Stuttgen and colleagues [81] summarized the frequency of germline changes of 30 genes in 100 mBC patients that varied by race/ethnicity and HR receptor status. A total of 14 patients harbored pathogenic variants, while an additional 21 patients harbored variants of unknown significance. Interestingly, 6% of patients displayed *BRCA1* variation, and key additional variants were found in *APC*, *ATM*, *BAP1*, *BARD1*, *BRCA2*, *BRIP1*, *CDH1* and *CHEK2*. Indeed, The Breast Cancer Association Consortium [82] interrogated rare germline variation of nine genes including *ATM*, *BARD1*, *BRCA1*, *BRCA2*, *CHEK2*, *PALB2*, *RAD51C*, *RAD51D* and *TP53* across 42,580 BC patients and 46,387 control subjects independent of family history of disease. Overall, the 9-gene panel was associated with over 27% of all early onset TNBCs. Specifically, *ATM* variation was strongest for HR+/HER2-negative patients with high tumor grade. *BARD1*, *BRCA1*, *PALB2*, *RAD51C* and *RAD51D* genetic variation was linked to TNBC overall risk, while *CHEK2* genetic variation was prevalent across all subgroups except for TNBC patients. *BRCA1* was associated with increased risk of all BC subtypes, with the highest overall risk for TNBC disease. *TP53* was associated with HER2-positive disease irrespective of HR status. Similarly, Couch et al. [83] evaluated associations of variation in *BRCA1/2* and 19 other genes in across a study population of over 41,000 Caucasian BC patients. In addition to variations of *BRCA1/2*, *CDH1*, *PTEN* and *TP53* that are highly associated with BC risk, variations in *ATM*, *BARD1*, *CHEK2*, *PALB2* and *RAD51D* were also associated with moderate or high BC risk. However, variants of *BRIP1*, *MLH1*, *MRE11A*, *NBN*, *NF1*, *PMS2*, *RAD50* and *RAD51C* were not linked to BC risk.

Characterizing germline variation of BC patients with varying ancestry shows interesting differences between populations. Han Chinese *BRCA1/2* mutation carriers identified by next generation sequencing account for 5% of BC patients studied. BC patients with *BRCA1/2* mutations are generally younger and are categorized into ER-negative and basal-like molecular subgroups [84]. Performing similar analyses on TNBC cases [85] showed that 16% of the cohort harbored at least one pathogenic variant and are associated with early onset disease. Interestingly, allele imbalances due to homologous recombination deficiency (HRD) in over 50% of cases, contributed by *BRCA1*, *BRCA1* and *RAD51D* mutations in 7.1%, 2.2% and 2.8% of cases, respectively. Indeed, *RAD51D* germline events were more common in Chinese TNBC patients than African American or Caucasian patients and are linked to aberrancies in HRD, DNA repair, as well as sensitivity to PARP inhibitors.

A survey of 1382 Greek BC patients with extensive family history of BC or with early onset disease revealed variation in nearly one-third of the cohort, with *BRCA1/2* loss of function variants found in nearly one in four cases [86]. This analysis also unveiled additional variants in *ATM*, *CHEK2*, *PALB2*, *RAD51C* and *TP53*. Similarly, germline analyses of early onset women from Latin America showed extensive alterations in 15% of patients [87]; however, a second report identified variants in 34% of patients studied, with *BRCA2* mutations in 4% of the cohort [88].

### 5.2. Somatic Mutation Profiles

The development and advancement of next-generation sequencing has enabled deep coverage of whole genome sequencing (WGS) and whole exome sequencing (WES) across large numbers of primary tumors, benign tissues, as well as apparently normal and normal-adjacent cells [89]. Primary human breast tumors have been extensively profiled using WGS and WES to identify somatic mutation events that are not only drivers of disease, but also potential efficacious targets for improving patient outcomes. In 2012, the Cancer Genome Atlas (TCGA) Network Consortium reported WES data for *n* = 510 primary invasive ductal BCs (IDCs), of which *n* = 466 primary tumors were also profiled for mRNA and miRNA gene expression, DNA methylation and copy number variation (CNV) [90]. Analysis of the mutation data [90] confirmed mutations in genes previously described that include *AKT1*, *CDH1*, *CDKN1B*, *GATA3*, *KTM2C*(*MLL3*), *MAP3K1*, *PIK3CA*, *PTEN*, *RB1* and *TP53*. This analyses also identified novel somatic mutations in several genes that include *CCND1*, *NF1*, *RUNX1*, *TBX3* and others [90]. Stratified by molecular subgroupings, somatic *PIK3CA* mutations were enriched in the Luminal A (45% of tumors), HER2 (39%) and Luminal B (29%) subtypes. *TP53* mutations were enriched in Basal-like (80%), HER2 (72%) and Luminal B (29%) subgroups. *GATA3* was altered in Luminal B (15%) and Luminal A (14%) subgroups, while *MAP3K1* was mutated in 13% of Luminal A tumors [90].

Stephens and colleagues from the International Cancer Genome Consortium [91] reported a separate WES analysis of 79 ER+ and 21 ER-negative BCs that confirmed the well-established somatic mutations in *AKT1*, *CDH1*, *GATA3*, *PIK3CA*, *PTEN*, *RB1* and *TP53*. Driver mutations in >40 genes were identified that includes *AKT2*, *ARID1B*, *CASP8*, *NCOR1* and *SMARCD1*, among others [91]. While the number of driver mutations did not correlate with tumor stage, the total number of driver and passenger mutations did positively correlate with tumor grade. In addition, the number of CpG to TpA substitutions in ER-negative BC patients correlated with patient age, whereas no correlation was identified in ER+ patients, suggesting that ER influences the mutation spectrum of BC patients [91].

In 2015, TCGA subsequently reported and compared multi-omic profiling data on *n* = 127 invasive lobular carcinomas (ILCs) to *n* = 490 IDCs and 88 mixed lobular and ductal tumors [92]. WES data for TCGA ILCs showed a high prevalence of mutations in *CDH1* (63%), *PIK3CA* (48%), *RUNX1* (10%), *TBX3* (9.5%) *PTEN* (7%) and *FOXA1* (7%) [92]. More recently, Pereira and colleagues performed DNA sequencing of the top 173 mutated genes across 2433 primary breast tumors [93]. As described in other reports, mutations in *PIK3CA* (40%) and *TP53* (35%) were the most abundant, while *MUC16* (17%), *AHNAK2* (16%) *SYNE1* (12%), *GATA3* (11%) and *KTM2C* (*MLL3*) (11%) mutations were also identified in at least 10% of all patients [93]. Interestingly, *BRCA1* and *BRCA2* somatic mutations are rare, with mutation frequencies <2% for both regions, while *TP53* germline mutations occurred in <1% of BCs [93]. These findings suggest that distinct profiles of germline and somatic mutations occur in driving BC tumorigenesis. By stratifying the somatic mutations by clinical tumor subgroup, ductal BCs displayed mostly *TP53* and PIK3CA mutations, lobular tumors displayed *CDH1* and *PIK3CA* mutations, and approximately 90% of medullary BCs harbored *TP53* mutations [93].

The Pereira report also revealed a set of 40 BC driver gene mutations [93], most of which were also identified by TCGA [90,92] and in the Catalogue of Somatic Mutations in Cancer (COSMIC) online database (https://cancer.sanger.ac.uk/cosmic, accessed on 5 June 2022). These driver mutations are in pathways involved in AKT signaling, cell cycle regulation, chromatin modifications, DNA damage, MAP kinase signaling, tissue organization, transcriptional regulation, and others [93]. Correlating these driver mutations to clinical BC patient outcome showed that functional: (1) *PIK3CA* mutations in ER+ tumors were linked to lower tumor grade, (2) *CBFB* and *GATA3* mutations were enriched for younger age patients, (3) *KTM2C/MLL* and *CDH1* mutations occurred in older patients and (4) *TP53* somatic mutations were linked to increased tumor grade irrespective of ER status [93].

Nik-Zainal and colleagues characterized WGS data for 560 primary BCs and identified 1628 protein-coding driver mutations in 93 genes [94], with at least one driver mutation in nearly every tumor analyzed. The most frequently mutated genes included *CCND1*, *ERBB2*, *FGFR1*, *GATA3*, *MAP3K1*, *MYC*, *PIK3CA*, *PTEN*, *RB1* and *TP53*. *TP53*, *MYC*, *PTEN* and *RB1* alterations were mostly found in ER-negative BCs, while mutations in *PIK3CA*, *CCND1*, *FGFR1* and *GATA3* were enriched in ER-positive BCs. Novel somatic alterations in *FOXP1*, *MED23*, *MLLT4*, *XBP1* and *ZFP36L1* were also unveiled. Inactivating germline or somatic *BRCA1* or *BRCA2* mutations were present in 16% of the cohort, and most of these cases also showed loss of the wild-type (WT) allele for these gene regions. As expected, *BRCA1* or *BRCA2* mutated BCs also included substantial genomic rearrangements that include small tandem duplications and deletions.

TCGA and the International Cancer Genome Consortium (ICGC) combined datasets to form the Pan-Cancer Analysis of Whole Genomes (PCAWG) Consortium, with the goal of summarizing genomic, epigenomic and transcriptomic variation for up to 6835 tumors across 38 cancer types [95]. With respect to BCs, PACWG showed that structural variation and point mutations are more prevalent in breast and ovarian cancers; however, BC point mutations contributed lower than colorectal cancer and B cell lymphomas [95]. Interestingly, chromothripsis, described by thousands of catastrophic mutations and rearrangements occurring in specific gene regions or chromosomes at one time, occurs most frequently in sarcomas, glioblastomas, squamous lung tumors, melanomas and breast cancers.

PCAWG also performed pathway analyses based on copy number and gene expression data from 1780 BCs stratified by molecular subgroups (basal-like, HER2, luminal A and luminal B). Pathway enrichment was most pronounced in apoptosis, chromosomal segregation, immune response and ribosomal biogenesis [96]. Specifically, apoptotic pathways were associated with patient prognosis in HER2 and Luminal A BCs. In this regard, *DUSP1* expression was the strongest apoptotic marker in HER2 BCs. DUSP1 is overexpressed in BCs and functions as a phosphatase signaling protein in suppressing apoptosis, thereby contributing to poor patient outcome. The immune pathway signature was associated with improved survival in basal-like and HER2 BCs, indicating that immune-based expression inhibits tumor progression, thereby improving patient outcome. Ribosome biogenesis pathways were almost exclusive to Luminal A BCs, while Luminal B tumors had enrichment in chromosome segregation.

### 5.3. Copy Number Variation in Breast Tumors

Somatic mutations may result in changes to protein activity, such as the inability to bind its target, misfunction, constitutive function or catalytic inactivation. However, most somatic mutations are mono-allelic; therefore, one WT copy can still rescue the mutant genotype. According to Knudson’s Two-hit hypothesis [97], complete gene inactivation requires both alleles to be targeted. This can be achieved by bi-allelic somatic mutation or a combination of mutation, deletion and/or silencing by promoter DNA hypermethylation [98]. Gene deletions and amplifications define the copy number variation (CNV) that is common across all human cancers [99]. Deletions of tumor suppressors or amplifications of oncogenes contribute to tumorigenesis and are influential in gene expression modulation. Technologies to identify changes include fluorescence in situ hybridization (FISH), comparative genome hybridization (CGH) and SNP arrays [100,101,102].

Focal and chromosomal arm level CNV changes are abundant in primary breast tumors. Breast tumors characteristically show *HER2*/*ERBB2* amplifications in the HER2 subgroup, chromosome (chr) 5q loss and chr10p amplifications in Basal-like BCs, chr 1q and chr 16q deletions in Luminal BCs [90]. In genes that displayed significant somatic mutations, TCGA identified focal amplifications in *EGFR*, *FOXA1*, *HER2* and *PIK3CA*; concurrently, this analysis showed focal deletions in *MAP2K4*, *KMT2C* (*MLL3*), *PTEN* and *RB1* [90]. While *HER2/ERBB2* amplifications were most substantially enriched in the HER2 subgroup, *HER2/ERBB2* amplifications were also evident in Luminal B tumors, and *HER2/ERBB2* deletions were seen in Basal-like BCs [90]. *PIK3CA* amplification was detected across all four molecular subgroups; however, this enrichment was strongest in Basal-like tumors. *MAP2K4*, *RB1* and *TP53* deletion events occurred in each subgroup, whereas *CDKN2A* and *KMT2C* deletions were more focused in Luminal B tumors. Arm-level amplifications in chromosome (chr) 1q, 7p, 8p, 8q, 16p, 17q, 20p and 20q occurred mostly in Luminal and HER2 tumors. Arm-deletions were substantial in the TCGA dataset, with these events occurring in most autosomal chromosomes [90].

The Pereira report included CNV and gene expression data on 2000 primary breast tumors from the METABRIC collection [93]. These CNV-driver events in which amplification or deletion affects gene expression levels include *CCND1*, *ERBB2*, *MYC*, *PAK* and *ZNF703*. Stratified by ER status, ER+ BCs displayed overwhelming gene deletions with *TP53*, *CHD1*, *CBFN*, *MAP2K4*, *PTEN* and *CTCF* as the most robust, with only *ERBB2* and *TBL1XR1* as amplified [93]. ER-negative BCs mostly showed *TP53* deletions, with lower deletion event frequencies in *RB1*, *CDH1*, *BAP1* and *PTEN*. ER-negative tumor amplifications were also focused on *ERBB2* with less frequent amplifications of *AKT1*, *CDKN1B*, *FOXO3*, *GATA3*, *KRAS*, *PIK3CA* and *TBL1XR1* [93].

### 5.4. Genetic Profiles of Metastatic Breast Tumors

Several reports have performed somatic mutation profiling of metastatic and non-metastatic breast tumors to identify genetic changes specific to metastases (summarized in [103]). While metastases are purportedly derived from the primary tumor, empirical evidence shows profound differences in somatic mutation profiles and frequencies [103]. Thus, identifying these metastasis-derived genomic alterations is important for improving mBC patient outcome. Roy-Chowdhuri and colleagues sequenced 46 hotspot genes for somatic mutations across 415 breast tumors, of which *n* = 110 were recurrent or metastatic [104] (Table 1); primary and metastatic lesions were analyzed for 61 BC patients. As stated in previous reports, somatic mutations in *TP53*, *PIK3CA*, *AKT1* and *ATM* were the most prevalent. TNBCs displayed the highest frequency of TP53 somatic mutations, while HER2 tumors displayed *TP53*, *PIK3CA* and *ATM* somatic mutations [104]. In comparing primary BCs to metastatic lesions, 77% of these pairs were identical. Most of the remaining pairs showed somatic mutations in the metastases that were not identified in the primary tumor, and the metastases also showed increased mutation frequencies, suggesting that these were acquired specifically in the metastatic process for genetic fitness and cell survival [105]. Other differences in mBCs versus non-metastatic tumors involve aberrant JAK/STAT signaling and SWI/SNF chromatin structure alterations in mBCs that may also serve as driver events [105].

Lefebvre et al. [106] generated WES data on 216 primary mBC—white blood cell pairs and compared the findings to non-metastatic mutation data from TCGA (Table 1). A total of 12 genes in descending order of mutation frequency (*AKT1*, *CBFB*, *CDH1*, *CDKN2A*, *ESR1*, *GATA3*, *MAP2K4*, *MAP3K1*, *PIK3CA*, *PTEN*, *RB1* and *TP53*) were significantly mutated in mBC over non-metastatic tumors, and eight additional genes (*AGR*, *EDC4*, *ESR1*, *FRAS1*, *FSIP2*, *IGFN1*, *OSBPL3* and *PALB2*) displayed higher mutation frequencies in mBCs than non-metastatic tumors. Interestingly, *ESR1* mutations were found in ER+ metastatic tumors from patients who were resistant to endocrine therapy, suggesting that *ESR1* mutations may be driver events in mBC progression. *ESR1* mutations reside in the ligand binding region and occur in up to 30% of mBCs, but are only detected at low frequencies in primary non-metastatic tumors. The most frequent *ESR1* mutations are gain-of-function alterations in the ligand-binding domain, resulting in constitutive enzyme activity in the absence of estrogen [107].

Ng et al. [108] performed multiregional sequencing on primary tumor and synchronous metastatic lesions in treatment-naïve patients with mBC to investigate their genomic profiles and revealed substantial difference with a median of 60% of shared somatic mutations (Table 1). Mutations were in known driver genes including, but not limited to, *GATA3*, *PI3KCA* and *TP53* and were preferentially clonal in both sites [108]. Mutations in genes participating in epithelial to mesenchymal transition, such as *SMAD4*, *TCF7L2* and *TCF4*, were either specific to or enriched in the metastasis.

Most driver mutations are clonal and occur on the trunk of a tumor’s phylogenetic tree [105]. Substantial evidence has supported that treatment pressure would result in the selection and the outgrowth of resistance subclones, contributing to treatment resistance [7,109]. Characterizing the genomic landscape of metastatic patients who have been treated will yield valuable insights into the effect of systemic treatment on the tumor genome and ultimately provide a basis for the development of novel therapeutic agents. Angus et al. [14] compared mutational profiles of 442 metastatic lesions to a well-characterized cohort of primary breast cancer (the BASIS cohort) and revealed that the median number of single-nucleotide variants (SNVs), insertions and deletions (Indels) and structural variants (SVs) were higher in metastatic lesions, consequently, leading to a higher tumor mutational burden (TMB), which doubles in the metastatic lesions (Table 1). TMB, a well-established predictive biomarker for immunotherapy, quantifies the total number of somatic coding mutations in a tumor. Furthermore, the mutation frequency of six well-known driver genes (*ESR1*, *TP53*, *NF1*, *AKT1*, *KMT2C* and *PTEN*) increases in metastatic lesions of ER+ breast cancer [14]. Conflicting data exist regarding the change of TMB in mBC. In contrast to the finding of Angus et al., one study [15] reported higher TMB only found in metastatic lesions of HR+ breast cancer and no difference in other subtypes (Table 1) and another demonstrated differential TMB in metastatic HR+ and triple negative breast cancer [110].

More recently, Rinaldi and colleagues [107] reported WES on over 11,000 primary breast tumors that include more than 5000 distal metastases. ER+ HER2-negative mBCs were more prevalent than in non-metastatic tumors; however, HER2 amplification frequencies were similar between BCs and mBCs. Somatic mutations in *TP53*, *PIK3CA*, *CDH1*, *GATA3*, *ESR1* and *KMT2D* were as previously described in other reports [14,90,104,106,108], while amplifications of *MYC*, *CCND1* and *HER2* and deletions of *PTEN*, *CDKN2A*, *CDKN2B* and *RB1* were the most frequent in non-metastatic breast tumors [107]. In addition to *ESR1* mutations in mBCs over non-metastatic tumors, *CTCF* driver mutations were discovered in 2% of mBCs, and amplification rates of the FGFR ligands *FGF3*, *FGF4* and *FGF9* were increased in mBCs.

Paul and colleagues [16] performed WES and WGS on 28 pairs of primary BCs and mBCs, as well as 38 unpaired mBCs, to identify somatic mutation and CNVs specific for mBCs (Table 1). Seven genes, *ESR1*, *EVC2*, *MYLK*, *PALB2*, *PEAK1* and *SLC2A4RG*, were preferentially mutated in mBCs and were not identified as mutated in TCGA BRCA tumors [16,90]. Comparing these genes to TCGA mutation data for over 30 tumor types showed that *ESR1*, *EVC2*, *MYLK*, *PEAK1* and *SLC2A4RG* were not significantly mutated in any other human cancer type [16]. CNV analyses showed metastatic-specific loss of *CDKN2A/B* and *STK11* concomitant with gains of *PTK6* and *PAQR8* [16]. Stratifying mutation, CNV and immunohistochemical data by pathways identified activation of mTOR, WNT and cAMP/PKA pathways and CDK/RB inactivation, as well as focal adhesion pathway changes [16]. Of note, mTOR signaling has been implicated as a mechanism of escape from therapies that target ER and HER2, as well as in clinical settings [111,112]. The preferential activation of mTOR in endocrine therapy-treated mBC patients provides a genomic basis for utilizing agents to inhibit the mTOR pathway for the treatment of mBC [111].

A meta-analysis of mutation and CNV data of 261 BC driver genes for 4268 mBCs and 5217 non-metastatic BCs [113] revealed that while most mutation frequencies were similar between mBC and non-mBC tumors, *ARID1A*, *ESR1* and *NF1* displayed significantly higher mutation frequencies in mBCs. Incorporating SNP-based analyses of insertions and deletions revealed mBC-enriched or mBC-specific alterations in *ARID1A*, *ATRX*, *AURKA*, *ESR1*, *FGFR4*, *NF1*, *PARP1*, *SMARCA4*, *STAG2* and *TSC2* [113]. Of these, *ESR1* displayed the highest frequency of amplifications and odds ratio of metastasis, and *ESR1* somatic mutations were enriched in liver metastases, but absent in positive lymph nodes. Interestingly, *TP53* mutations were enriched in positive lymph nodes, but absent in liver metastases [113].

### 5.5. Chromosomal Instability (CIN)

Chromosomal abnormalities have been reported in tumor progression in several cancers, including breast cancer [20]. Chromosomal instability (CIN), results from errors in chromosome segregation during mitosis, is a hallmark of cancer, and has been proposed to act as a driver in metastasis. CIN equips tumors with enhanced evolutionary capabilities and facilitates the development of treatment resistance by generating intra-tumoral heterogeneity [114]. WGS analyses have demonstrated that cancer genomes acquire CIN to evade oncogene addiction [115]. CIN is a robust feature in several cancers, including mBC [116]. Numerous studies have demonstrated a poorer treatment and/or survival outcome of patients with higher CIN in multiple cancer types, including breast cancer [115,117,118].

Analysis of a pan-cancer cohort (Memorial Sloan Kettering-Metastatic Events and Tropisms) consisting of more than 25,000 metastatic patients with tumor profiling and clinical information revealed that metastatic lesions were more chromosomally unstable than primary tumors for several cancers including HR+ mBC, evident by a higher fraction of genome altered (FGA) and a higher TMB [119]. They further investigated the clinical significance of TMB by comparing the percentage of patients with TMB high (≥10 mutations/Mb) metastatic tumors vs. primary tumors. They showed that a significantly higher percentage of TMB high tumors were observed in metastases of HR + HER2- patients [119]. Furthermore, the frequency of whole-genome duplication (WGD) was higher in the metastatic tumor of HR + HER2- compared to primary tumors [119]. The major differences between primary and metastatic lesions are summarized in Figure 2.

Collectively, studies on genomic landscapes and chromosomal aberrations have demonstrated that although metastases are clonally related to the primary tumor, evident by the sharing of many driver mutations, they are biologically distinct from the primary tumors, evident by increased genomic instability, the presence of unique mutations and harboring a specific HR or HER2 status, highlighting the need to treat metastases as a distinct clinical and biological entity.

## 6. DNA Methylation Alterations in Breast Cancer

In addition to somatic mutations and copy number alterations driving carcinogenesis, epigenetic mechanisms are important regulators of gene expression in human cancers. While somatic mutations alter gene product function, epigenetic changes modulate gene expression levels [98]. Epigenetic modifier genes acquire somatic mutations in human cancer, and the epigenetic mechanisms of DNA methylation, chromatin modifications, nucleosome positioning are also early and ubiquitous across nearly all forms of human cancer [120]. The most well studied epigenetic mechanism is DNA methylation, the covalent modification of the C-5 position of cytosines in a 5′-CpG-3′ (CpG) DNA sequence context [121]. Human cancers generally display global DNA hypomethylation that is represented by repetitive element and CpG poor DNA hypomethylation [122]. Alternately, CpG rich promoter regions, termed promoter CpG islands, are generally unmethylated in normal somatic cells, but display DNA hypermethylation in cancers that may be associated with reduced gene expression levels [122]. Gene body regions are methylated in actively transcribed genes, and gene body DNA hypomethylation correlates with reduced gene activity [123]. The presence of DNA methylation marks may serve as beacons for the recruitment of co-repressors to silence genes or co-activators; concurrently, the absence of DNA methylation may allow for the recruitment of transcription factors to activate gene expression [122].

DNA methylation changes do not occur in isolation, but correlate with modifications (methylation, acetylation, phosphorylation, etc.) of specific amino acid residues on the N-terminal tails of histones H2A, H2B, H3 and H4 [124]. Since genomic DNA wraps around the histone core octamer, histone modifications are instrumental in opening or closing chromatin to regulate gene expression [124]. For example, Histone H3 lysine 4 trimethylation (H3K4me3) is located at promoters of actively transcribed genes, while H3K9me3 is a marker of heterochromatin and silencing of repetitive elements [124]. H3K36me3 is found in gene bodies of actively expressed genes [124]. In addition, H3K27 acetylation (H3K27Ac) is a marker of active gene expression, however, H3K27me3 is linked to silenced genes [122]. H3K4me1 is associated with enhancers that distally control gene expression, while chromatin regions with both H3K4me3 and H3K27me3 marks are associated with poised enhancers [124]. Both DNA methylation and histone modifications co-exist with nucleosome positioning in regulating gene expression [125]. Actively expressed genes are noted by the absence of nucleosomes to allow RNA polymerase and related transcription machinery. Alternatively, repressed genes show highly compacted nucleosomes that ultimately block the access of transcription machinery to these regions [125].

Human BC is a rich source of cancer-specific epigenetic alterations. Interestingly, while *BRCA1* germline mutations are frequent in BC, *BRCA1* somatic mutations are much less frequent. Notably, BCs display loss of heterozygosity (LOH) of *BRCA1* as well as reduced *BRCA1* mRNA expression due to promoter DNA methylation of the remaining allele [126,127,128,129,130]. In addition to *BRCA1* gene regulation, ERα expression is an important facet of breast tumor biology and patient outcome. ERα expression is based in the activity of the *ESR1* gene, which is unmethylated in normal breast epithelium that correlates with active ERα expression [131].

Human primary breast tumors display *ESR1* promoter DNA hypermethylation that is inversely correlated with *ESR1* gene expression. ESR1 DNA hypermethylation correlates with poor patient outcome and correlates with response to tamoxifen [132,133,134,135]. A subset (37%) of BC patients with *ESR1* promoter DNA hypermethylation developed loss of ER expression as a function of metastasis. Primary tumors in which *ESR1* was unmethylated never displayed ER expression loss in metastases [131], implicating *ESR1* DNA methylation as a potential driver of breast cancer metastasis.

In addition to *BRCA1* and *ESR1* driver events, DNA hypermethylation changes in BC occur in genes that are involved in key cellular processes [136,137,138] that include: (1) apoptotic regulation—*APC*, *BCL2*, *DAPK*, *DCC*, *HOXA5*, *HIC1*, *TWIST* and *TMS1*; (2) cell cycle regulation—*CCND2*, *CDH1*, *CDKN2A*, *FOXA2*, *SFRP1*, *WIF1*, *WRN*, WT1 and *SFN (14-3-3σ)*; (3) cellular homeostasis—*GPC3*, *HOXD11*, *LAMA3*, *LAMB3*, *LAMC2* and *ROBO1*; (4) DNA repair—*ATM*, *BRCA1*, *MGMT* and *MLH1*; (5) hormone and receptor signaling—*ESR1*, *PGR*, *RARB2*, *RASSF1*; (6) inhibition of angiogenesis—*SFRP5*, *THBS1*; and (7) negative regulation of tumor invasion and metastasis—*CDH1*, *CDH13*, *CST6*, *SYK*, *TIMP3*. Concurrently, DNA hypomethylation of genes involved in positively modulating metastasis and tumor aggressiveness include *BCSG1*, *CDH3*, *NAT1* and UPA [137,138]. de Ruijter et al. [139] performed a meta-analysis of DNA methylation data on 87 loci across 72 reports to identify biomarkers correlative to patient outcome. DNA hypermethylation of *RASSF1*, *BRCA1*, *PITX2*, *CDH1*, *RARB*, *PCDH10*, *GSTP1* and *PGR* was associated with poor patient outcome [139]. Comparing DCIS to invasive ductal carcinoma [140] has revealed increased DNA methylation frequencies of *APC*, *CACNA1A*, *CDH1*, *HOXA10*, *MGMT*, *TRFAP2A* and *TWIST1* in IDC over DCIS specimens. *CDH1*, *MGMT*, and *SFRP1* displayed increased DNA methylation levels in IDC over DCIS. Stratifying DCIS by HR status revealed DNA hypermethylation of *ABCB1*, *FOXC1*, *GSTP1*, and *RASSF1* in ER+ DCIS. *GSTP1* also showed increased DNA methylation levels in PR+ DCIS [140].

Identifying aberrant DNA methylation profiles in human cancers has been propelled due to technological advancements, and the most paramount is the use of bisulfite to delineate between methylated and unmethylated cytosines (reviewed in [141]). Bisulfite treatment of genomic DNA converts unmethylated cytosines to uracil (replaced by thymine in PCR applications), while methylated cytosines are refractory to treatment; thus, generating a DNA sequence change that can be exploited using PCR and sequencing technologies [141]. Interrogating individual loci by PCR and methylation-sensitive restriction enzyme methods provided only a limited picture of cancer methylomes, and profiling DNA methylation levels on a genome-scale (microarrays) [142,143,144] or in a genome-wide manner (whole genome bisulfite sequencing (WGBS)) [142,145,146,147] has allowed for the discovery of DNA methylation changes across the genome. DNA methylation microarrays have been used on large numbers of primary tumors by TCGA [90,92,148] and other groups [149,150,151] to identify patient subgroups, correlate DNA methylation events to patient outcome and response to treatment, as well as comparisons to mutation and CNV profiles.

Hill and colleagues [150] generated Illumina DNA methylation array data on 39 BCs and four tumor/normal-adjacent pairs and compared these data to clinical features of breast cancer patients. Unsupervised two-dimensional clustering of selected probes categorized BCs into three groups: Group 1 tumors displayed high DNA methylation and are associated with relapse and are positive for ER and PGR. Alternatively, Group 3 tumors exhibited low DNA methylation and were enriched for TNBCs. Probe-based analyses identified six genes (*ACADL*, *ITR*, *RECK*, *SFRP2*, *UAP1L1* and *UGT3A1*) in which promoter DNA hypermethylation of each gene was associated with disease relapse and shortened survival after diagnosis [150].

A separate promoter DNA methylation array analysis of 14,000 genes across 236 infiltrating BCs [151] showed similar clustering based on ER status; however, the unmethylated group clustered with the ER-enriched subgroup. Clustering of the entire tumor collection yielded six DNA methylation subgroups, in which HER2-positive tumors were enriched in Group 2, basal-like tumors in Group 3 and Luminal A tumors in Group 6 [151]. The remaining groups were mixes of expression subgroups: Group 1 contained HER2, basal-like and Luminal B tumors, Group 4 contained HER2 and Luminal B tumors, and Group 5 contained Luminal A and B tumors [151]. Comparison of DNA methylation and gene expression subgroup genes revealed that the DNA methylation signatures reflect the T cell lymphocytic infiltration of tumor tissues, implicating not only the tumor microenvironment, but also an immune component reflected in the DNA methylation data [151].

TCGA reported five BC subgroups based on the unsupervised clustering of Illumina DNA methylation array data of gene promoters and gene bodies for all genes in the human genome across 466 breast tumors [90]. Of note, Group 2 and Group 5 tumors were generally HER2-negative. Group 5 tumors were identified by the lack of cancer-specific DNA methylation an overlapped with Basal-like expression subgrouping, extensive *TP53* mutations, and the absence of ER and PGR expression [90]. Group 3 tumors displayed a DNA hypermethylated phenotype that was enriched for Luminal B expression subgrouping, the absence of *MAP2K4*, *MAP3K21* and *PIK3CA* mutations [90]. A separate analysis by Fang and colleagues [148] identified DNA hypermethylated tumors as belonging to a CpG island methylator phenotype (CIMP), defined as cancer-specific DNA hypermethylation of a subset of CpG islands in a subset of tumors. CIMP was first presented in 1999 for colorectal cancer [152] and CIMPs have been subsequently identified in endometrial, brain and gastric tumors (reviewed in [153]). Breast-CIMP (B-CIMP) tumors display a DNA methylation profile that determines metastatic potential. B-CIMP was associated with low risk of metastasis and improved patient outcome indicating that DNA hypomethylation is correlated with metastasis in breast cancer [148].

A comparison of promoter DNA methylation profiles between inflammatory BCs (IBCs) and non-IBCs identified only four regions with differential DNA methylation: *AGT*, *TJP3*, *MOGAT2* and *NTSR2* [149]. All four loci displayed DNA hypermethylation in IBCs as compared to non-IBCs. Unsupervised clustering analyses of the entire data set revealed three DNA methylation subgroups: normal-like, high DNA methylation and low DNA methylation [149]. Tumors in the high-DNA methylation group were metastatic and these patients had poor outcome [149]. Pathway analyses of genes displaying differential DNA methylation between the low and high DNA methylation subgroups showed differences in genes related to chemokine signaling, cytokine receptor activation, focal adhesion, Wnt signaling and metabolism of pyruvate, galactose, fructose and mannose [149]. Overall, the differential DNA methylation between IBCs and non-IBCs indicates the potential role of DNA methylation in metastasis and subsequent poor patient outcome.

## 7. Therapeutic Options for Breast Cancer Patients

Current established BC treatment options include chemotherapy (CT), endocrine therapy (ET), radiation therapy (RT), as well as targeted therapies (reviewed in [55,154]), which are selected based on HR and HER2 status. ER+/PGR+/HER2-negative BC patients mostly receive endocrine therapies such as tamoxifen, letrozole, anastrozole or exemestane; however, cytotoxic chemotherapies such as adriamycin (Doxorubicin), cyclophosphamide, paclitaxel, docetaxel or carboplatin are administered in selected patients. Tamoxifen functions to inhibit estrogen binding to ER to modulate ER function, while letrozole, anastrozole and exemestane are aromatase inhibitors that inhibit the conversion of androgens to estrogen. Cyclophosphamide is a DNA alkylating agent that complicates DNA replication, adriamycin is a DNA intercalator that interferes with topoisomerase-mediated DNA repair, docetaxel and paclitaxel inhibit microtubule function to disrupt mitosis and carboplatin functions by facilitating DNA crosslinks to inhibit DNA replication [55]. HER2+ patients usually receive chemotherapies as well as the HER2-targeted monoclonal antibody therapies trastuzumab (Herceptin) or pertuzumab [55]. If the HER2+ BC patient is HR+, then endocrine therapies are administered. Finally, TNBC patients mostly receive chemotherapies that include adriamycin/cyclophosphamide, adriamycin/cyclophosphamide/paclitaxel or docetaxel/cyclophosphamide.

## 8. Established Treatment Schemes for Metastatic Breast Cancer Patients

Historically, mBC was treated with the assumption that key biological features were shared between primary and metastatic lesions. However, in-depth analyses of primary and metastatic lesions of breast cancer have revealed substantial differences in genomic landscapes and significantly advanced our understanding of the effects of systemic treatments on the tumor genome, and ultimately broadens the treatment options for mBC patients. Cancer cells seeding metastases continue to evolve by acquiring novel somatic and epigenetic alterations. However, the role of genomic alterations in driving the metastatic process is not well-understood.

Therapeutic options for mBC patients are also guided by HR and HER2 status [55]. ER+/PR+/HER2– mBC patients receive CDK4/6 and aromatase inhibitors as first line therapy [55]. Subsequent lines of therapy involve hormone-based therapies such as tamoxifen or targeted therapies based on menopause status. If the mBC patient also has *BRCA1* or *BRCA2* germline mutations, the PARP inhibitors olaparib or talazoparib are administered. Subsequent lines of therapy are usually single-agent cytotoxic chemotherapies [55]. HER2+ mBC patients receive (1) taxane, Herceptin and pertuzumab as first line therapy; (2) endocrine therapy with HER2-based therapeutics if the patient is ER+/PGR+/HER2+, or (3) ado-trastuzumab emtansine, which is herceptin covalently linked to mertansine (DM1) [55]. DM1 inhibits microtubule assembly by interacting with tubulin to inhibit DNA replication and cell division. Subsequent therapy lines for HER2+ mBC patients involve Herceptin with chemotherapy or endocrine therapy if the tumor is ER+/PGR+. Finally, metastatic TNBC (mTNBC) patients initially receive single-agent chemotherapies including taxane, platinum or anthracycline; however, the response rates for these agents are <40% with PFS of 3–5 months [55]. Subsequent lines of therapy for mTNBC patients include single agent drugs such as capecitabine (a prodrug that is metabolized to 5-fluorouracil to inhibit DNA replication), eribulin (inhibits microtubule assembly to inhibit cell division), gemcitabine (a cytidine analog that inhibits DNA replication) or vinorelbine (inhibits mitosis) [55].

## 9. Novel Molecular Therapeutic Targets for mBC

Tumoral heterogeneity is observed at morphological, phenotypic, and molecular levels and represents a significant clinical challenge in accurately characterizing and treating the disease. Human cancers undergo considerable molecular and cellular evolution under therapeutic pressure as well as during development, giving rise to spatial and temporal heterogeneity. Clonal evolution supports the notion that treatment exerts pressure leading to the selection and the outgrowth of resistant subclones [7]. Unfortunately, several metastatic-specific driver mutations are related to treatment resistance rather than metastatic process [155]. mBC patient survival is dismal, since secondary lesions are often refractory to conventional therapies [155]; therefore, it is necessary to develop novel treatments for mBC.

Given the effects of *BRCA1/2* alterations on DNA repair and homologous recombination in BC, treatments directed towards DNA repair are now recognized as promising targeting strategies for these patients. Poly(ADP-ribose) polymerases (PARPs) bind to DNA strand breaks as an early response to DNA damage and recruit machinery to repair the lesion, thereby promoting cell survival. Treating cancer cells with PARP inhibitors (PARPi) blocks the ability of PARP to initiate DNA repair, thus leading to accumulated DNA damage and ultimately, apoptosis. PARPi have been studied as single- and dual-agents in treating BC patients (reviewed in [156,157]). Recent meta-analyses [157,158] of clinical trials utilizing PARPi for BC patients showed efficacy in improving OS, as well as improved PFS for TNBC, HR-positive, and or HER2-negative patients, as well as those with *BRCA1/2* germline mutations. PARPi showed improvement in tumor drug response, OS and PFS for patients with locally advanced or metastatic HER2-negative BC with *BRCA1/2* germline variation [158], thus supporting the use of PARPi in BC patient care.

There is strong evidence for genome evolution of HR+ HER2^−^ breast cancer, which counts for approximately 70% of all breast cancer cases, from its early stage to metastatic stage. Many of the metastasis-enriched driver mutations are a result of resistance to treatment. Mutations in *RB1*, a tumor suppressor regulating the G_1_-S phase transition through transcriptional repression of *E2F1* [159], are frequently found in patients treated with CDK4/6 inhibitors [160], which in combination with an aromatase inhibitor, have become the preferred frontline therapy for endocrine-sensitive metastatic HR+/HER2^−^ breast cancer [161]. This treatment scheme showed significant improvements in both PFS and OS observed in four clinical trials: PALOMA-2, MONARCH-3, MONALEESA-2 and MONALEESA-7 [161]. The MONALEESA trials have demonstrated that patients with *RB1* mutation at baseline (pre-treatment) had a significantly shorter PFS (3.7 months), suggesting that *RB1* mutation represents a primary resistance to CDK4/6 inhibition when combined with endocrine therapies [162]. Therefore, targeting *RB1* deficiency has significant therapeutic potential. Preclinical data have demonstrated that RB1-decificent cells are dependent on Aurora B kinase for survival, and inhibition of Aurora kinase is synthetic lethal [163]. Therefore, Aurora kinase inhibitors are potentially efficacious for patients with RB1-mutant tumors who progressed on CDK4/6 inhibitors. In addition, preclinical studies have highlighted that a triple combination of endocrine therapy, CDK4/6 inhibition and PI3K inhibition prevents CDK4/6 inhibitor resistance [164]. In vitro studies have shown that ER+ breast cancer cells quickly adapt to CDK4/6 inhibition, which can be prevented by co-treatment with a PI3K inhibitor. Triple combination treatments in patient-derived tumor xerograph (PDX) models resulted in rapid tumor regression, resulting in superior efficacy than the double treatment scheme [165].

Multiple studies have shown that mutations in *NF1*, a negative regulator of Ras signaling, have been frequently observed in HR+/HER2–mBC patients and importantly, *NF1* mutations were mutually exclusive with *ESR1* mutations in this patient group [166]. *NF1* mutations lead to the activation of the RAS/MAPK pathway [166]. Patients with *NF1* somatic mutations had poor response to hormonal therapies, suggesting an attenuated ER dependence and resistance to endocrine therapy [15]. NF1 has been the focus of therapeutic intervention, the most popular and promising strategy is RAS protein targeting [167]. There is potential therapeutic efficacy in blocking critical protein–protein interactions between receptor tyrosine kinase (RTK) and RAS activation, such as GRB2, SOS or SHP2 [168]. Furthermore, NF1 loss results in sustained MAPK pathway activation, and is rescued by ERK inhibition, suggesting a potential therapeutic option [167,169,170].

The mTOR pathway has also been implicated as a mechanism of escape from therapies targeting ER and HER2 in vivo. *MTOR* is hypermutated in mBC patients who have been treated with endocrine therapy or had liver metastases [111,112]. Paul et al. identified that *MTOR* hypermutation occurs regardless of ER-therapy exposure or liver metastases, providing a genomic basis for the use of mTOR inhibitors in mBC patients [16]. In addition to somatic mutations, some studies have shown that oncogenic amplification or activation may represent a common mode of breast cancer evolution under treatment pressure [105]. Amplifications of *MDM4*, *FGFR1* and *CCND1* were observed in patients treated with endocrine therapy [105] and may also serve as potential therapeutic targets.

Epigenetic therapy, namely inhibition of DNA methylation, is a promising therapeutic agent in treating cancer patients. 5-aza-2′-deoxycytidine (5-Aza-CdR, Decitabine, DAC) and 5-azacytidine (5-Aza-CR, Vidaza, AZA) are FDA-approved for MDS [171,172,173], but DNA methylation inhibitor treatments of solid tumors as single agents have been inadequate, and may be explained by stability and toxicity of the compound to limited impact on the treatment. However, there are several new DNA methylation inhibitors, such as GSK3685032 recently published by Pappalardi et al., that show much longer half-lives and reduced toxicity compared to DAC and AZA [174], and may have better outcomes for clinical application, especially for patients with solid tumors [175,176,177]. DNA methylation inhibitors have anti-neoplastic effects including reactivation of tumor suppressor genes [121,178], down-regulation of oncogenes [123], reactivation of transposable elements (TE) to stimulate a “viral mimicry” immune response [179,180] and re-sensitization to chemotherapies [181] such as poly(ADP-ribose) Polymerase (PARP) inhibitors (PARPi) [182] and RNA splicing inhibitors [183] that hold clinical promise in treating *BRCA1*-mutated BCs.

## 10. Concluding Remarks

Scientific and clinical advancements have resulted in a substantial improvement in breast cancer morbidity and mortality. Although the advancements in the treatments for mBC have resulted in improvements in progression-free survival and/or overall survival, drug resistance inevitably rises in almost all patients. A comprehensive understanding of the genomic landscape of breast cancer metastasis could reveal the clinical impact of tumoral heterogeneity as well as facilitate the development of novel therapeutic agents. This review provides a comprehensive view on the genomic landscape of mBC and highlights potentially novel targets for therapeutic development.

## Figures and Tables

**Figure 1 genes-13-01555-f001:**
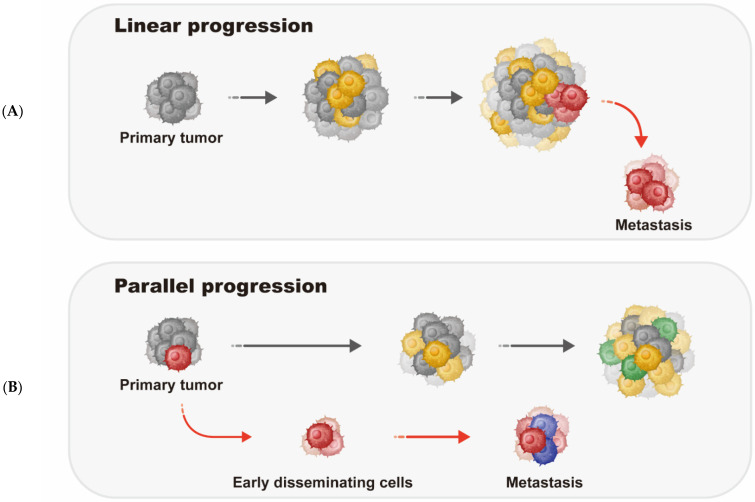
Models of cancer metastasis. Two models of cancer metastasis have been proposed: the linear model (**A**) and the parallel model (**B**). In the linear model, metastatic clones arise late in the evolution of the primary tumor when they are equipped with all the necessary properties to disseminate and successfully colonize a secondary site (**A**). In contrast, in the parallel model, metastatic clones escape from the primary tumor early and evolve separately and eventually forming metastasis (**B**).

**Figure 2 genes-13-01555-f002:**
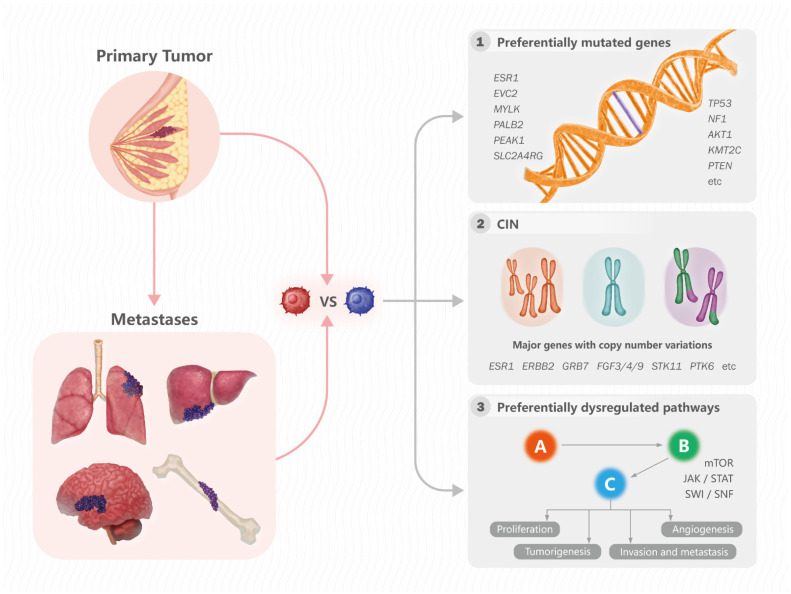
Major differences between primary and metastatic lesions. Major differences between primary and metastatic lesions in somatic mutations, chromosomal instability and molecular pathways. Major genes with copy number variations are also listed.

**Table 1 genes-13-01555-t001:** Studies on the genomic landscape of metastatic breast cancer.

Study	Patients	Samples	Subtypes	Sequencing Approach
Roy-Chowdhuri et al. [104]	354	305 primary 110 recurrent/metastatic	224 ER/PR+ HER2- 33 ER/PR+ HER2+ 8 ER/PR- HER2+ 89 ER/PR- HER2-	A 46 cancer-related gene panel
Yates et al. [105]	170	148 locoregional 79 metastatic 72 unknown	87 ER+/HER2- 34 HER2+ 37 TNBC 12 unknown	A 365 gene panel
Lefebvre et al. [106]	216	Paired metastatic and blood samples	143 HR+/HER2- 51 HR-/HER2- 14 HER2+	WES
Ng et al. [108]	9	Paired primary and synchronous metastatic samples	3 HR+/HER2+ 2 HR+/HER2- 2 HR-/HER2+ 2 TNBC	WES
Angus et al. [14]	442	Metastatic samples	279 ER+/HER2- 49 ER+/HER2+ 28 ER-/HER2+ 58 TNBC 28 unknown	WES
Bertucci et al. [15]	617	543 metastatic samples 74 breast tumors	381 ER+/HER2- 30 HER2+ 182 TNBC 24 unknown	WES
Paul et al. [16]	66	28 paired primary and metastatic samples 38 unpaired	3 HR+/HER2+ 43 HR+/HER2- 5 HR-/HER2+ 12 TNBC 3 unknown	WES and WGS

ER: estrogen receptor; PR: progesterone receptor; HER2: human epidermal growth factor receptor 2; TNBC: triple negative breast cancer; WES: whole exon sequencing; WGS: whole genome sequencing.

## Data Availability

Not Applicable.

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
