# Peer review of "The Multi-Omic Landscape of Primary Breast Tumors and Their Metastases: Expanding the Efficacy of Actionable Therapeutic Targets"

_genes, 2022, doi:10.3390/genes13091555_

Round 1

Reviewer 1 Report

Major points need to be addressed:

Fig. 1. Elaborating what criteria or molecular characteristics were used to define linear and parallel progression in both text and figure legend.

Fig. 2. Specifying major mBC-enriched alterations in copy number of genes/chromosomal regions in figure and legend

Table 1 is missing

It will be more informative to tabulate the following information: 1) What drugable genes were associated with metastasis; 2) what are the underlying causes of the altered expression of these genes in mBC; 3) what drugs are available or in clinical trials to target these genes

The following statements need to be clarified, elaborated or referenced:

·         “Cancer cells ARE NOT UNDER POSITIVE SELECTION PRESSURE to metastasize; rather, SELECTED HALLMARKS, including increased…” It is needed to elaborate how the “SELECTED HALLMARKS” evolve during tumor progression.

·     These two statements appear to be contradictory: “It should be noted, however, that LINEAR AND PARALLEL MODELS OF METASTASIS DO NOT ACCOUNT FOR LYMPH NODE INVOLVEMENT” vs. “In agreement with the Kroigard report (Kroigard et al., 2017), BOTH LINEAR AND PARALLEL PROGRESSION MODELS OF METASTASIS WERE EVIDENT” .

·      These two statements appear to be contradictory: “Tumor-positive lymph nodes are common in cancer metastasis and PROVIDES CONTEXT FOR METASTATIC TUMOR-CELL SEEDING AND TRANSPORT TO DISTAL TISSUES VIA THE LYMPH SYSTEM in addition to the bloodstream (Ullah et al., 2018)” vs “Most notably, THE ANALYSES ALSO SHOWED THAT METASTATIC LYMPH NODES ARE NOT INVOLVED IN METASTATIC SEEDING, suggesting that mBC is achieved through angiogenesis and CTC spread via the bloodstream”

·         Defining “chemokine cells”.

·         "TNBCs are an aggressive form of human breast cancer, with high rates of metastasis, tumor recurrence, poor overall patient survival and INCREASING INCIDENCE RATES (of what?) over time.”

·         “These are poorly understood tumors that account for 5-10% of all BCs and resemble Luminal A BCs”.  Elaborating how does normal-like BC resemble Luminal A BC (e.g., diagnostic IHC markers, gene expression patterns and prognosis) and providing references.

·         It is difficult to understand this statement: “Since TNBCs also show similar gene expression profiles to basal-like BCs, the majority (80%) of TNBCs are indeed basal-like, but are thought to be more homogeneous than TNBCs”.

·         While somatic mutations alter gene product function, epigenetic changes MODULATE ACTIVITY LEVELS (of what?).

·         Correcting typo in this statement: “while BRCA1 germline mutations are IN FREQUENT in BC, BRCA1 somatic mutations are much less frequent”

·         It is difficult to follow the logic of this sentence: “No primary tumors in which ESR1 was unmethylated ultimately showed ER loss in metastases, implicating ESR1 DNA methylation as a potential driver of breast cancer metastasis”

·         “Probe based analyses identified six genes (ACADL, ITR, RECK, SFRP2, UAP1L1 and UGT3A1) that are significantly associated with relapse-free survival”. Elaborating how the methylation status (e.g. methylation intensity and position) of these genes are linked to BC relapse

·         Specifying what genome regions (e.g, CpG island in promoter regions vs. whole genome) were examined by Hill et al. 2011, Dedeurwaerder et al., 2011, and TCGA. In conclusive/summary statements, it is necessary to specify Methylation status in what genome regions were linked to BC prognosis

Reviewer 2 Report

In their review paper with the title “The multi-omic landscape of metastatic breast cancer: expanding the efficacy of actionable therapeutic targets”, Yang et al. summarize the differential molecular landscapes, the genomic heterogeneity and the therapeutic relevant features of breast cancer and metastatic breast cancer.

The authors have made a great effort to extensively summarize the literature. However, I have some comments that I think will strengthen their review.

1.       Even though the title refers to metastatic BC, I feel that the review was on BC, with just a small focus on mBC. That doesn’t take away from the review’s value, but maybe authors should re-consider the title or the paper’s structure.

2.       The reference list is not extensive. This is a review paper, so almost each sentence should have its own citation. For example, in lines 569-581 there is a whole paragraph without any citation! Some other examples are lines 32-35, 68-69, 72-80, 86-95, 133-135, 138-139, 191-194. These are only some examples, so the authors should make sure that there is an adequate number of citations throughout the whole manuscript.

I understand that some things can be basic knowledge for the authors, but for a scientist with a different background, such as a student or a computational biologist, everything should be supported with references.

Moreover, I think that many paragraphs were a summary of a single paper. This is appropriate in some cases, but in other cases, for instance when statistics are presented, the authors should have multiple sources to crosscheck.

3.       Continuing with the references, in some parts of the manuscript the references are very old.

For example, in the germline variants subparagraph, the newest reference is a decade old. This is a scientific field that has seen tremendous novelty with the introduction of Next Generation Sequencing and especially with the introduction of gene panel sequencing in the clinical diagnostics ca. 2016.

Examples of papers that could be referenced included (and are not limited to):

a.       Breast Cancer Association Consortium article in Jama Oncol 2022

b.       Ding Ma et al. JNCI: Journal of the National Cancer Institute 2021

c.       Couch et al Jama Oncol 2017

d.       Stuttgen et al, Jama Oncol 2019 (This is even for metastatic BC!)

And maybe even some population-based papers:           

e.       Fostira et al, Journal of Medical Genetics, 2019

f.        Guoli et al, Journal of Cancer Research and Clinical Oncology 2017

g.       Gómez-Flores-Ramos Cancers 2022

Similarly, the list of citations for the somatic variants is not up-to-date. For example, the authors should include relevant articles from the series of PCAWG paper in Nature in 2020. Another paper that extensively studies BC somatic genomic landscape is Nik-Zainal et al, Nature 2016.

On the opposite side, in the Molecular Subgroups of Breast Cancer sub-section, Charles M. Perou et al. Nature, 2000 is missing. Even though it’s old, this is the most important paper that established the molecular subtypes.

4.       Authors should mention a new emerging subtype that seems to respond differently to treatment, mainly HER2-low BC. The first papers published on this subtype were as late as 2020.

5.       Authors should discuss more PARP inhibitor treatment (only briefly mentioned in treatment for metastatic BC sub-section), as well as cisplatin-based chemotherapy for BRCA mutation carriers.

6.       English language needs improvement. I noticed many typos, grammatical and syntax errors.

Round 2

Reviewer 1 Report

NA

Reviewer 2 Report

First, I'd like to apologize to the authors for my late reply due to a technical problem.

The authors followed both my and the other reviewer's suggestion thoroughly. I feel that after the additions and corrections, this review now summarizes the literature on BC in great detail. 

I also like the new title much better.

I suggest acceptance of the manuscript in present form.